# Mechanism Analysis of Transcription Factor OsERF110 Regulating Rice Pollen Response to Heavy Ion Irradiation

**DOI:** 10.3390/biology14091218

**Published:** 2025-09-08

**Authors:** Kai Sun, Jinzhao Liu, Jiameng Zhang, Haonan Li, Jian Zeng, Libin Zhou, Tao Guo, Chun Chen

**Affiliations:** 1National Engineering Research Center of Plant Space Breeding, South China Agricultural University, Guangzhou 510642, China; sunkai1994@scau.edu.cn (K.S.); ljz999999@stu.scau.edu.cn (J.L.); 15137007082@163.com (J.Z.); 20251015018@stu.scau.edu.cn (H.L.); 2School of Biology and Agriculture, Shaoguan University, Shaoguan 512005, China; zengjian@sgu.edu.cn; 3College of Agriculture, South China Agricultural University, Guangzhou 510642, China; 4Institute of Modern Physics, Chinese Academy of Sciences, Lanzhou 730030, China; libinzhou@impcas.ac.cn

**Keywords:** rice, pollen, heavy ion irradiation, transcriptional regulation

## Abstract

Heavy ion irradiation, as an efficient physical mutagenesis technology, exhibits important applications in crop breeding, but its damage mechanism and regulatory network of rice pollen are still unclear. As a model crop, rice pollen has become an ideal material for studying the mechanism of radiation response because of its single-cell characteristics, radiation sensitivity, and direct transmission of genetic damage. In this study, we systematically analyzed the dynamic process of DNA damage and repair in rice pollen under heavy ion irradiation and determined that the transcription factor OsERF110 regulates the DNA repair, oxidative stress, and metabolic reprogramming network in rice pollen to deal with heavy ion irradiation damage. This provides a new perspective for the target mining of crop radiation breeding and the study of the germ cell DNA repair mechanism. The purpose of this study was to reveal the molecular mechanism of rice pollen response to heavy ion irradiation and to explore the key regulatory genes to provide a theoretical basis for crop radiation mutation breeding.

## 1. Introduction

Heavy ions refer to the positively charged atomic nuclei remaining after nitrogen, carbon, and other atoms are stripped or partially stripped of their peripheral electrons [1]. The energetic rays formed by accelerating heavy ions through a large accelerator device are heavy ion beams. As a physical mutagenesis technology, heavy ion irradiation displays significant advantages over traditional methods (such as X-ray or γ-ray) due to its unique physical and biological characteristics (such as high let value, dense damage, etc.). Compared with low linear energy transfer (LET) radiation (such as protons or photons), heavy ion beams produce stronger biological effects in biological tissues, such as higher cell killing efficiency and a unique damage mode. Therefore, heavy ion beams can more effectively cause DNA damage and significantly increase the mutation frequency, thus inducing more extensive phenotypic variation [2,3,4,5].

Heavy ion irradiation has been applied to the breeding and mutagenesis mechanism research of many species [6]. Traditional heavy ion mutagenesis of rice mainly focuses on somatic tissues such as seeds. Germ cells (such as pollen) are usually more sensitive to radiation than are somatic cells. Low-dose irradiation may stimulate pollen viability due to “hormesis”, while high-dose irradiation may induce a higher mutation rate than that of somatic cells, resulting in fertility decline [7,8,9,10]. At the same time, as a typical self pollination sexual reproduction crop, DNA damage and genome variation in gametes of rice can be directly inherited to offspring through fertilization. In addition, as a single-cell system, pollen avoids the interference of tissue heterogeneity, is conducive to the precise study of the radiation damage mechanism at the cellular level, and also provides a unique model for the study of double fertilization and mutation heritability [11].

DNA is the main target of heavy ion radiation damage, especially considering that DNA double-strand breaks (DSBs) are one of the most serious damage types caused by radiation, and its effective repair is essential for cell survival and genome stability [12,13]. DSBs in plants are mainly repaired through two core pathways, namely homologous recombination (HR) and non-homologous end joining (NHEJ). HR is a precise repair mechanism that relies on homologous templates such as sister chromatids for repair. NHEJ is an efficient but error prone repair mechanism that directly connects the broken ends, without relying on homologous templates [14,15,16]. However, these studies mainly focus on seeds, seedlings, or somatic cells, with relatively less research on pollen or other reproductive cells. Although the DNA damage response (DDR) mechanism in plant somatic cells has been preliminarily studied, the specific repair pathway in germ cells is still unclear [17,18]. For example, some studies have suggested that plant germ cells may coordinate DDR through specific transcription factors (such as SOG1), but the regulatory differences between germ cells and somatic cells have not been clarified [19,20]. Research on Arabidopsis root tip cells has shown that RAD51 transcriptional response exhibits cell type and developmental region specificity, but it is unknown whether germ cells follow the same pattern [21]. At present, there have been few reports on the mechanism of DNA damage repair in plant germ cells, such as the HSP101 protein encoded by the *MS42* gene in maize pollen mother cells, which plays a key role in RAD51 loading, DSB repair, and meiosis [22]. The function of OsMLH1 during meiosis in rice and its mutants was found to exhibit a semi-sterile phenotype [23]. OsMLH1 and OsMLH3 synergistically affect the exchange of homologous chromosomes during meiosis of rice gametes, which in turn affects pollen fertility [24]. In addition, although it is known that the transcriptional upregulation of cell cycle checkpoints, DNA repair genes, and replication related genes (such as RAD51, BRCA1, RPA1E, etc.) is involved in DSB repair in plants, there is still a lack of research on the repair dynamics, gene expression, and regulatory mechanisms of DSBs in pollen or other germ cells.

In this study, rice pollen was treated by heavy ion irradiation, combined with molecular biology experiments, RNA sequencing (RNA-seq), and DNA affinity purification sequencing (DAP-seq), to reveal the effect of heavy ion irradiation on the survival rate of rice pollen and the repair of DSBs, to analyze the temporal changes of gene expression and key pathways after irradiation, and to identify the core transcription factor OsERF110 and its target gene network that regulate this process. The results of this study can provide new insights into the molecular mechanism of plant germ cell response to heavy ion irradiation, providing a theoretical basis for the key gene mining for crop radiation mutation breeding.

## 2. Materials and Methods

### 2.1. Heavy Ion Irradiation

The material used in this study was japonica rice pure line variety Nipponbare, which was planted under standardized conditions in the teaching and research base of South China Agricultural University in Guangzhou (SCAU), Guangdong Province. At the 6–7 stage of young panicle differentiation, spikelets at the appropriate stage were selected for heavy ion irradiation. The high-energy ion beam ^12^C^6+^ irradiation, provided by the Lanzhou Heavy Ion Research Facility of the Institute of Modern Physics, Chinese Academy of Sciences, was used. The irradiation energy was 80.55 MeV/u, and the dose rate was 1 Gy/min. A total of four dose gradients of 2, 4, 8, and 16 Gy were set. Each dose was repeated in three dishes, and the unirradiated samples were prepared as the control.

### 2.2. Preparation of Pollen Cell Suspension

The anthers in spikelets were separated on the sterile operation table, and 1 × phosphate buffer saline (PBS) buffer was dripped and gently squeezed to release the pollen. The pollen was collected in a 3.5 cm diameter Petri dish, an appropriate amount of 1 × PBS buffer was added, and the density was adjusted to 10^4^/mL; the dish was then sealed for aseptic storage.

### 2.3. Pollen Activity Detection

Take an appropriate amount of pollen cell suspension and identify the pollen activity via the fluorescein diacetate–propidium iodide (FDA–PI) fluorescence staining method. Under the fluorescence microscope, the living cells are green, and the dead cells are red. The FDA stock solution was diluted 1000 times with 1 × PBS, and the PI stock solution was diluted 2000 times. The diluted FDA and PI were mixed in a 1:1 ration to prepare the FDA–PI working solution. Take 1 mL of cell suspension from each Petri dish and put it into a 1.5 mL centrifuge tube. Centrifuge for 10 min at 1000 r/min; discard the supernatant and add 200 μL FDA–PI working solution. Stain at room temperature for 30 min, rinse with 1 × PBS buffer three times, and then resuspend. Each tube was used to extract 50 μL of cell suspension for observation under a fluorescent microscope. This process was repeated three times, and 10 fields were observed in each repetition. Each field was photographed and recorded.

### 2.4. Immunofluorescence Experiment

Spikelets at the 6–7 stage of young panicle differentiation were irradiated by heavy ions and fixed by adding 4% paraformaldehyde at 0, 1, 2, 6, and 12 h after irradiation. The irradiated samples were set as the control. First, the fixed spikelets were embedded in paraffin, and then the wax blocks were sliced with a slicer with a thickness of 5 μm, dewaxed with xylene, and washed with water. Sodium citrate was used for antigen repair. A 3% H_2_O_2_ solution prepared with 30% H_2_O_2_ and methanol was added to each slice, followed by incubation at room temperature for 10 min to block the activity of endogenous peroxidase. Slices were blocked with 3% bovine serum albumin (BSA) for 30 min, the blocking solution was removed, 50 μL of γ-H2AX primary antibody was added to each slice, and the slices were held overnight at 4 °C. The slices were washed with PBS three times, followed by the addition of secondary antibody, and incubation at room temperature for 2 h. The slices were then washed with PBS three times, and 4,6-diamino-2-phenyl indole (DAPI) (1 μg/mL) was added, followed by incubation at room temperature for 10 min. The slices were then washed with PBS three times and incubated with autofluorescence quenching agent for 5 min. After washing, an anti-fluorescence quenching agent was added to seal the film. Finally, the slices were imaged via a 3DH fluorescence scanner.

### 2.5. RNA-Seq

At 0, 1, and 6 h after irradiation, an appropriate amount of pollen cell suspension was extracted. The corresponding numbers of samples in the treatment group were C1, C2, and C3, and those in the control group were CK1, CK2, and CK3. Each sample was tested three times. The total RNA was extracted using Omega Plant RNA kit (Omega Bio TEK, Norcross, GA, USA, R6827), and the RNA quality and integrity were detected by Qubit (Thermo Fisher Scientific, Waltham, MA, USA) and Agilent 2100 (Agilent Technologies, San Jose, CA, USA). The qualified samples were constructed and sequenced by Illumina.

First, we use Fastp (v0.20.0) to perform quality control on the raw reads downloaded from the machine, filtering out low-quality data, including removing reads containing adapters, reads with an N ratio greater than 10%, reads that were all A bases, and reads with a quality value Q ≤ 20 that account for more than 50% of the entire read in order to obtain clean reads. The process is as follows: Use the short reads alignment tool Bowtie2 (v2.5.4) to align clean reads to the ribosome database, remove reads from the aligned ribosomes, and use the remaining unmapped reads for subsequent transcriptome analysis. Conduct comparative analysis based on the reference genome using HISAT (v2.1.0), including type statistics, gene coverage, sequencing randomness, and sequencing saturation analysis. Reconstruct transcripts using Stringtie (v1.3.1) and utilize RSEM (http://deweylab.Giyhub.Io/RESM/, accessed on 5 May 2025) to calculate the expression levels of all genes in each sample, display them as fragments per kill of exon model per million mapped fragments (FPKM), and use FPKM as a screening indicator. Perform differential analysis using edgeR (v4.0), where | log_2_FC | > 1 and FDR < 0.05 output differentially expressed genes (DEGs).

We utilize R (http://www.r-project.org/, accessed on 9 May 2025) to conduct principal component analysis (PCA) to study the distance relationship between samples using dimensionality reduction techniques, as follows: Transfer the target gene to the Gene Ontology (GO) database (http://www.geneontology.org/, accessed on 18 May 2025) to obtain the GO term with significantly enriched genes. Integrate the target gene set with the Kyoto Encyclopedia of Genes and Genomes (KEGG) database (https://www.kegg.jp/, accessed on 19 May 2025). By combining this method with enrichment analysis, the pathways with significant enrichment were screened. Mfuzz clusters based on the Fuzzy C-Means Clustering (FCM) algorithm, two-group trend analysis was performed on the treatment group and the control group. The goal is to calculate the different multiple gene expressions between the treatment trend group and the corresponding control trend group, screen the genes with multiple differences greater than 2 in at least one trend group, and finally take the log_2_ value of the differences between multiples for trend analysis.

### 2.6. WGCNA

The weighted gene co-expression network analysis (WGCNA) algorithm first assumes that the gene network follows a scale-free network distribution, where the logarithm of the number of connected nodes (log (i)) is negatively correlated with the logarithm of the probability of this node appearing (log (p (i))). Build a WGCNA network using the R language package; first, calculate the expression correlation coefficient between genes, and then find the power value that makes the data conform to the scale-free distribution as a whole. Construct a gene clustering tree with power = 8, and divide the gene modules based on the clustering relationship between genes. Genes with similar expression patterns will be grouped into the same module, and the branches of the clustering tree will be cut and distinguished to generate different modules, with each color representing a module. After preliminary module partitioning, we obtain the Dynamic Tree Cut of the preliminary partitioned modules. As some modules are very similar, we will merge modules with similar expression patterns based on the similarity of module feature values to obtain the final partitioned module Merged Dynamic. The similarity selected for this analysis is 0.6. Display the expression patterns of module genes in various samples using module feature values, and draw a heatmap of sample expression patterns, reflecting the comprehensive expression levels of all genes in the module in each sample.

### 2.7. Subcellular Localization

In order to construct the 35S:OsERF110-GFP fusion expression vector, the coding sequence (CDS) of *OsERF110* was cloned from tobacco by using the vector pEXT06G1, and the plasmid pEXT06G1-OsERF110 was recombined between BamH I and Spe I sites. After activation of the Agrobacterium plate, fresh single colonies were selected and added to Rif-resistant LB, and the colonies were cultured at 230 r/min and 28 °C for 24 h. The bacterial solution was expanded and cultured. A total of 200 μL 0.5 M/L MES and 4 μL 100 mM/L were added to the new LB with corresponding antibiotics, and then 67 μL Agrobacterium solution was inoculated and cultured to OD600 = 0.5–0.6. The process is as follows: Collect the bacteria by centrifugation, suspend the bacteria with the infection solution, measure the OD600, then extract the calculated amount of bacteria solution into a 5 mL centrifuge tube, dilute it to 4 mL with the infection solution, and evenly mix multiple target bacteria solutions in equal proportion. The syringe extracts the bacterial solution, injects it into the back of tobacco (*Nicotiana benthamiana*) leaves, and marks the position. Tobacco was treated in dark for 12 h after injection and then cultured for 48 h. Fluorescence imaging was performed via laser confocal microscopy.

The same method was used to construct the rice p35S:OsERF110-GFP fusion expression vector. About 5 g of the stems and leaves of rice seedlings grown for 10 days were cut with a blade, and 10 mL of enzymatic hydrolysate (1.5% Cellulose RS, 0.03% Pectinase Y23, 0.5 M Mannitol, 0.5 mM KCl, 0.5 mM MgCl, 0.5 mM, pH 5.7 MES, 10 mM CaCl, and 0.1% BSA) were added. The protoplasts were collected by filtration and centrifugation, washed with W5 solution, and resuspended with MMG solution for 5 h. Then, take 100 μL of protoplast suspension, add 10 μL of DNA (plasmid DNA purity is greater than 500 ng/μL, with the addition of 10 μL of co-transfection marker), and an equal volume of PEG4000 solution, mix well, and let stand at room temperature for 30 min. Add 1 mL of W5 solution for dilution, and mix well to terminate the reaction. The protoplasts were collected by centrifugation, and the supernatant was discarded. The protoplasts were cultured at 28 °C for 24 h in the dark. The protoplasts were observed by laser confocal microscopy, and the images were output.

### 2.8. DAP-Seq

Take an equal amount of pollen cell suspension from each sample and mix. Use the NEXTFLEX Rapid DNA Seq Kit (PerkinElmer, Waltham, MA, USA, NOVA-5188-02) to extract gDNA from the pollen, and construct the fragments of gDNA into a library. Clone the CDS of OsERF110 into the pFN19K HaloTag T7 SP6 Flexi vector, and express it using the TNT SP6 coupled wheat germ extraction system (Promega, Madison, WI, USA, L4130). Purify and capture the expressed proteins using Magne Halo Tag Beads (Promega, Madison, WI, USA, G7281). Incubate the gDNA library linked to the OsERF110 binding beads and the adapters together. After incubation, the DNA was broken into short fragments using ultrasound technology, followed by end repair and 3′ end addition, and connected to Illumina sequencing adapters. DNA fragments of 100–300 bp were selected for PCR amplification to obtain a qualified library for sequencing. Sequencing of eluted DNA was performed using two replicated techniques on the Illumina NavoSeq6000. Without adding protein to the beads, use it as input for the negative control DAP library. The offline data were compared to the reference genome using Bowtie2 (V2.2.8), and the unique reads were used for subsequent analysis. Use deepTools (V3.2.0) to count the reads, set a window with a size of 50 bp, and calculate the average reads depth in each window. Using MACS2 (V2.1.2) to conduct peak calling in the whole genome, the threshold is a q-value < 0.05. The position information of the peak in the genome and the sequence information of the peak region were analyzed, and the peak-related genes were screened. Then, conduct intra-group peak merging and output unified peak data. Using the R package of CHIPseeker (V1.16.1), the peak related genes were annotated.

## 3. Results

### 3.1. Survival Rate of Pollen Irradiated by Different Doses

In order to obtain more mutations and maintain pollen cell activity, to a certain extent, the median lethal dose (LD50) was set as the optimal dose for subsequent studies. In this study, the pollen activity after different doses of heavy ion irradiation was identified (Figure 1A–E), the survival rate was calculated, and the dose survival rate regression equation was constructed (Figure 1F). With the increase in irradiation dose, the pollen survival rate after heavy ion irradiation showed a significant downward trend, in which the pollen survival rate of 2 Gy was 72.05%, the survival rate of 4 Gy was 44.77%, the survival rate of 8 Gy was 12.76%, and the survival rate of 16 Gy was only 3.22%. The regression equation of the dose–survival relationship was y = 100.82e^−0^·^222x^ (R^2^ = 0.9858). This equation shows a good fit, and the calculated theoretical LD50 was 3.16 Gy. According to the above theoretical LD50 value, the dose closest to the theoretical value was selected as the actual sampling dose in this study, i.e., 4 Gy was used as the LD50 for subsequent research.

### 3.2. DSBs Repair at Different Time Points After Irradiation

Histone H2AX plays an important role in the repair of DSBs. Once DSBs are produced, it phosphorylates at ser139 site and becomes γ-H2AX, which can mark the site of DSBs. In this study, the repair of DSBs after heavy ion irradiation was observed by immunofluorescence at five time points of 0, 1, 2, 6, and 12 h after irradiation. At the end of irradiation, i.e., 0 h, there were a large number of green fluorescent spots in the pollen cells, indicating that heavy ion irradiation triggered a large number of DSBs, which had not been repaired. At 1 h after irradiation, the number of DSBs decreased, indicating that the repair process of DSBs in pollen cells was continuing. At 2 h after irradiation, most DSBs had been repaired, until 6 h after irradiation, DSBs had been basically repaired. Only a small number of large green fluorescent spots (complex damage) existed, and complex damage still existed until 12 h after irradiation (Figure 2). These results demonstrate that most pollen DSBs induced by heavy ion irradiation could be repaired rapidly, while some more complex lesions required at least 12 h to repair.

### 3.3. Gene Expression Differences at Different Time Points After Irradiation

Based on the results of immunofluorescence, RNA-seq was performed at 0, 1, and 6 h after irradiation. According to the principal components analysis (PCA) results of the samples, the clusters between the repeated samples were relatively concentrated, the expression patterns were relatively consistent, and the clusters between different regions were obviously separated, indicating that heavy ion irradiation had a significant impact on the gene expression patterns of pollen cells, and there were differences in the expression patterns at different time points after irradiation (Figure 3A). After screening, a total of 5556 DEGs were obtained (Appendix A). There was partial overlap among the DEGs at the three sampling time points, and the number of upregulated DEGs was lower than that of downregulated DEGs. At the same time, with the delayed sampling time, the number of DEGs gradually decreased (Figure 3B,C). In order to further classify the DEGs according to the expression pattern, the trend analysis was carried out. The DEGs of the treatment group and the control group at the same time point were divided into 10 clusters according to the differential expression of the genes. We focused on cluster 1, cluster 4, and cluster 10. A total of 801 genes were included in these three clusters, and their expression and differential expression showed a continuous downward trend with the extension of sampling time, while the genes in the other seven clusters did not show a continuous change trend (Figure 3D).

To clarify the function of the DEGs, GO and KEGG enrichment analyses were performed, respectively. The results of GO enrichment showed that during 0–1 h after irradiation, the functions of DEGs were relatively similar, mostly related to the response to external stimuli (such as temperature, reactive oxygen species, etc.) and protein folding and binding, and most of them were located in ribosomes and cytoplasm, indicating that pollen cells were in the process of synthesizing related proteins in response to irradiation stimulation. At 6 h after irradiation, the function of the DEGs was mainly related to the synthesis of a variety of organic substances (such as lipids), as well as enzyme and protein activities, and most of them were related to the cell membrane (Figure 4A). From the above enrichment results, it can be seen that when the pollen cells were only affected by irradiation, the main biological activities of the cells were to cope with stress and repair damage. After 6 h, the damage repair was basically completed, and the main function turned to the metabolic processes of normal life activity. In the enrichment results of KEGG, DEGs at three time points simultaneously enriched protein processing, plant pathogen interaction, the plant MAPK signaling pathway, and the synthesis and metabolism of a variety of organic substances (such as fatty acids, diterpenes, biotin, glutathione, phenylpropanoid, etc.) in the endoplasmic reticulum. There were multiple pathways related to the response to abiotic stress damage, such as phenylpropanoid biosynthesis, the plant MAPK signaling pathway, and glutathione resistance to reactive oxygen species damage, which were also mutually confirmed with the GO enriched pathway (Figure 4B), reflecting the way pollen cells respond to radiation damage.

### 3.4. Co Expression Network Analysis of Radiation Responsive Genes

To further clarify the differences in gene expression patterns, 21,902 genes (FPKM > 2) detected by RNA-seq were analyzed by WGCNA. Under the condition of power = 8, 21,902 genes were divided into 19 modules (module 1–19) according to the differences in expression patterns. The maximum module was module 3, with 5836 genes, and the minimum module was module 5, with only 1 gene. We focused on module 19, which was highly expressed in the samples of the treatment group. With the delay in sampling time, the expression showed a continuous downward trend (Figure 5A,B). There were 1403 genes in module 19 which were enriched by GO and KEGG. The results of GO enrichment showed that these were related to the response to a variety of stimuli (such as temperature, reactive oxygen species, etc.) and the protein folding process (Figure 5C). KEGG was significantly enriched in protein processing, a variety of DNA damage repair pathways (such as HR, NHEJ, etc.), and the synthesis and metabolism of a variety of organic substances (such as glutathione, terpenes, sugars, flavonoids, etc.) in the endoplasmic reticulum (Figure 5D). The enrichment results of module 19 were similar to those of the DEGs, which confirmed that module 19 selected in this study was closely related to the gene response process of heavy ion irradiation.

### 3.5. Identification of Key Genes in Response to Heavy Ion Irradiation

In order to mine the key response genes in response to heavy ion irradiation, firstly, 1403 genes in module 19 were intersected with 801 genes in trend analysis cluster 1, cluster 4, and cluster 10. The weight value in WGCNA reflects the correlation between gene pairs. In order to reduce the complexity of the co expression network, minimize background noise, and preserve almost all biologically significant strong connections, focusing on the most core and reliable co expression relationships, we selected the top 10,000 gene pairs with weight values from the crossover genes. WGCNA module connectivity reflects the importance of genes in the module, and then the top 100 connectivity genes of module 19 were selected. Finally, the gene pairs screened above were mapped into a coexpression network (Figure 6A). According to the coexpression network, *Os02g0546600* was at the core of the coexpression network and was associated with the largest number of genes (Figure 6B). Therefore, we believed that *Os02g0546600* played an important role in the response of rice pollen to heavy ion irradiation.

### 3.6. Structure and Subcellular Localization of OsERF110

*Os02g0546600* is 756 bp long, including two exons and one intron. CDS is 654 bp long, of which the second exon is 425 bp long, encoding a highly conserved AP2/ERF domain (Figure 7A), that is, Os02g0546600 encodes a transcription factor in the AP2/ERF family, so it is named OsERF110. In order to clarify the role of OsERF110 in cells, the CDS sequence of *OsERF110* was fused with GFP, was transiently expressed in rice protoplasts and tobacco leaves, and its subcellular localization was observed. Confocal laser microscopy of tobacco leaves and rice protoplasts showed that OsERF110-GFP fusion protein was specifically distributed in the nucleus and cell membrane (Figure 7B,C). These results indicate that OsERF110 is located in the nucleus and cell membrane, which is consistent with the functional characteristics of OsERF110 as a transcription factor involved in DNA transcription initiation, regulation, and response signal pathway, and is also consistent with the enrichment results of GO, a cellular component of DEGs, indicating that there may be a potential regulatory relationship between OsERF110 and DEGs.

### 3.7. Identification of Target Genes of Transcription Factor OsERF110

OsERF110, as a transcription factor, usually regulates the expression of downstream target genes by binding to specific DNA sequences. Therefore, DAP-seq was carried out on OsERF110 in order to identify the DNA sequence bound to OsERF110 and the target genes regulated by OsERF110. A total of 65,538 and 60,032 peaks were detected in the two biological replicates of DAP-seq, of which 45,680 peaks were detected repeatedly (Figure 8A), and the consistency of the two biological replicates was 69.7% and 76.09%, respectively. The 45,680 peaks were mostly distributed at the transcription starting site (TSS), and the number gradually decreased with the increase in distance, showing a normal distribution (Figure 8B), which was consistent with the characteristics of DAP-seq peaks. The above results showed the reliability of the DAP-seq results.

From the distribution of peaks in the genome, the proportion of peaks located in the intergenic region was the greatest, up to 30.42%, followed by the results for the promoter region. The proportion of peaks within 2 kb of the promoter accounted for 19.61%, and the number of peaks was the least, accounting for only 2.39%. The number of peaks distributed in the genome from high to low was intergenic > promoter > intron > exon > downstream > 3′ UTR > 5′ UTR (Figure 8C).

The binding sequence of transcription factors is usually located in the promoter region of downstream target genes. Therefore, we focused on the target genes of peaks located in the promoter region, with a total of 8302 (Appendix A). GO and KEGG enrichment were performed on these. The results of GO enrichment showed that the target genes were significantly correlated with DNA repair, response to stimulation, and synthesis and metabolism of a variety of organic compounds (Figure 8D). KEGG enrichment results showed that the target genes were involved in DNA damage repair (NHEJ), a variety of organic compounds, synthesis and metabolism (such as fatty acids), TCA cycle, and basic transcription factors (Figure 8E). The enrichment results of target genes were partially consistent with those of DEGs and WGCNA, which verified the authenticity, functional relevance, and synergy of OsERF110 regulation.

### 3.8. Analysis of OsERF110 Regulatory Network

In order to further screen and clarify the target genes and regulatory network of OsERF110, we carried out a joint analysis of RNA-seq and DAP-seq. First, the 8302 target genes with peaks located in the promoter region of DAP-seq and 5836 DEGs of RNA-seq were intersected, with a total of 1098 genes (Figure 9A). The number and characteristics of the intersecting genes showed a similar trend to those of DEGs; that is, with the extension of sampling time, the number of genes gradually decreased, and the number of genes with downregulated expression was more than that with upregulated expression (Figure 9B).

Among the 10 clusters in the trend analysis, we found that *OsERF110* was located in cluster 4. Therefore, we speculated that the expression levels and different expression multiples of the target genes regulated by OsERF110 should also showed a similar trend with *OsERF110*. Further, the intersection gene and the cluster 4 gene were intersected. Finally, we got a total of 62 reliable target genes of OsERF110 (Appendix A), and their expression levels are as shown in Figure 9C. The expression trend of 62 target genes and *OsERF110* was consistent; they were highly expressed in the treatment group, and the expression level was the highest in sample C1 and the lowest in sample C3.

The functions of 62 target genes were annotated and classified to clarify the regulatory network of OsERF110. The results showed that OsERF110 of rice pollen responded to heavy ion radiation by activating gene expression of multiple pathways and biological processes. Aminoacyl tRNA biosynthesis, RNA modification, and the time regulation of the transition from a vegetative stage to a reproductive stage were only specifically expressed during C1, which also displayed the least number of genes, reflecting the timeliness of the gene response of these three pathways to heavy ion irradiation. The pathways responsible for stimulation, the protein process, DNA repair, and the synthesis and metabolism of a variety of organic substances (such as fatty acids, phenylpropanoid, etc.) were the pathways containing the most genes, and their genes were specifically expressed during C1 and C2, indicating that within 1 h after heavy ion exposure, OsERF110 displayed the most active stage in regulating the expression of related response genes, and also the most critical stage in activating the response pathway to repair DNA damage. The plant MAPK signaling pathway, ion transport, the redox process, synthesis and metabolism of a variety of organic substances (such as terpenes, steroids, etc.), and other pathway genes were continuously expressed in C1, C2, and C3 within 6 h of irradiation, indicating that although the radiation damage had been basically repaired at this time, the redox process caused by irradiation required a long time to adjust, and the gene expression of pollen cells tended to return to the process of normal life activity after 6 h of irradiation (Figure 9D).

## 4. Discussion

This study identified the core transcription factor OsERF110 that responds to heavy ion irradiation in rice pollen and revealed its molecular network that synergistically responds to radiation damage by regulating DNA repair, stress response, and metabolic pathways. Plant heavy ion irradiation materials are mostly somatic tissues such as seeds and seedlings, among which seeds, as representatives of somatic cells, usually exhibit high radiation tolerance and require high dose levels to cause significant growth inhibition or mutation. For example, the semi-lethal dose of plant somatic cells such as rice, hyacinth, and upland cotton can reach tens or even hundreds of Gy [25,26,27,28]. The LD50 of rice pollen in this study was 3.16 Gy, which is much lower than the reported LD50 of somatic cells. Moreover, when rice pollen was irradiated with heavy ions of 16 Gy, its survival rate was only 3.22%. The traditional view is that heavy ion mutagenesis requires high doses to induce significant damage, but the results of this study indicate that germ cells exhibit “low-dose high sensitivity”, suggesting that dose strategies can be optimized for germ cells in radiation mutagenesis breeding. Pollen, as a male gametophyte, highly relies on a fine ion dynamic balance to maintain polar growth and fertilization processes. However, heavy ion irradiation directly induces an outbreak of reactive oxygen species (ROS), disrupting ion homeostasis and leading to cellular dysfunction and even death [8]. In contrast, the metabolic activity of seeds stagnates, and the repair mechanism is more easily activated, which may reduce acute sensitivity to radiation [29]. In addition, pollen cells display limited DNA repair ability during meiosis, and their high LET characteristics may make them more susceptible to damage [30]. On the contrary, seeds display stronger DNA repair ability, a lower metabolic rate, and a more stable structure, thereby enhancing tolerance [31]. Therefore, pollen, made up of highly differentiated single cells, relies on a fine ion balance and active metabolism for its function. Heavy ion irradiation directly disrupts this homeostasis, making pollen more susceptible to death.

The repair speed of DSBs induced by high LET heavy ions is significantly slower than for those exposed to low LET radiation (such as protons and helium ions), and this delay is mainly due to the more complex structure of DSBs caused by heavy ions (such as cluster like breaks), resulting in a decrease in the efficiency of the repair pathway [32]. About 70–80% of simple DSBs undergo rapid initiation of repair processes in the early stages (0–1 h) after irradiation, with repair proteins (such as γ-H2AX, ATM) rapidly recruited at individual injury sites [33,34]. In the mid stage after irradiation (1–2 h), the kinetics begin to differentiate, and the repair rate of high LET heavy ion irradiation is lower than that of low LET irradiation [35]. In the late stage after irradiation (24 h), the repair effect is poor, and residual damage is common, which also reflects the long-term threat of heavy ion irradiation to genome stability [36,37,38]. This study set five time points after rice pollen heavy ion irradiation and demonstrated the dynamic repair process of DSBs through γ-H2AX. The results showed that most DSBs can be quickly repaired in a short period of time, while some complex injuries require at least 12 h of repair time, and the repair process is accompanied by temporal activation of OsERF110 and its target genes (such as RAD51 in the HR and KU70 in the NHEJ). This result is also consistent with previous reports, intuitively demonstrating the difficulty of repairing “clustered DNA damage” caused by the high LET characteristics of heavy ion beams. The residual γ-H2AX signal may indicate complex damage with delayed repair or cells heading towards apoptosis. The delayed heavy ion repair phenomenon observed in human cell lines has been confirmed in this study in the unique system of plant germ cells, providing a phenotypic basis for its subsequent transcriptional regulation mechanisms. In addition, contrary to the traditional understanding that “NHEJ is the main repair pathway” in somatic cells [14,15,16], this study also found that the HR and NHEJ in pollen cells are coregulated, suggesting that this is a specific mechanism evolved by germ cells to ensure genome stability and reduce heritable mutations.

After radiation-induced DNA damage in plants, complex regulatory networks will be activated to maintain genome integrity, which is called DDR. The functional diversity and integrity of DDR genes and mechanisms are the determinants of maintaining genome stability [39]. The DDR mechanism of plants displays different emphasis for different response times, showing a time gradient. For example, in the early stage (<2 h), the DDR pathway was rapidly activated, and the number of differentially expressed genes reached the peak, mainly involving DNA damage response, RNA splicing, deadenylation, RNA destabilization, and other pathways [40], and the expression of a large number of proteins (such as proteasome subunits) was upregulated, accelerating the degradation of misfolded proteins [41]. The results of this study also support this view. The number of DEGs is the largest at 0 h after irradiation, and the number is the least at 6 h. with the extension of time, the number gradually decreases. DEGs are widely involved in the process of protein folding and also significantly enriched in the related pathways and processes of stimulus response. At the same time, OsERF110 also responds to this process by upregulating the expression of genes related to DNA damage repair and RNA modification. In the middle and late stage (>2 h), gene expression is mainly the synergistic effect of cycle arrest, antioxidants, and radiation-induced ROS burst; plants respond by upregulating antioxidant and metabolic pathways (such as glycolysis and oxidative phosphorylation) [5]; TCA cycle and antioxidant genes are also rapidly upregulated, resisting oxidative stress and enhancing energy metabolism to support the repair process; and MAPK signaling pathway genes are continuously activated [42]. This study found that at 2–6 h after irradiation, DEGs were also significantly enriched in redox, glucose metabolism, and TCA-related pathways to maintain the DDR process, and the MAPK signaling pathway, ion transport, flavonoids, terpenes, and fatty acid synthesis and metabolism related pathways were continuously activated during this period. This result reflects the three-stage dynamic model of “emergency–repair–recovery” of pollen cells in response to irradiation. Unlike previous studies, this study fully depicts this dynamic process in pollen through precise time point tracking and finds that there may be differences in timing and pathway emphasis between it and somatic cell response. In the early stage, it displays comprehensive defense and repair mobilization, while in the later stage, the focus shifts to clearing side effects and restoring normal life activities.

Transcription factors, also known as trans-acting factors, are protein molecules that can specifically bind to cis-acting elements in the promoter region of eukaryotic genes. Transcription factors interact with cis-acting elements of downstream target genes to regulate the expression of target genes. At present, more than 60 transcription factors have been identified in plants, such as AP2/ERF, MYB, NAC, WRKY, etc., which play an important role in plant growth, development, and environmental response [43]. Transcription factors have been repeatedly confirmed to play an important role in the DDR process of plants. For example, SOG1 plays a key regulatory role in plants similar to that of P53 in animals, responsible for responding to DNA damage and activating downstream genes to coordinate cell cycle arrest, DNA repair, programmed cell death, and other processes [19,20]. MdWRKY72 responds to UVB radiation in apple and promotes pigment accumulation by activating anthocyanin synthase genes (such as pal and ufgt) [44]. E2Fa and E2Fb regulate the inhibition of cell proliferation under UVB radiation and affect plant growth [45]. Radiation sensitivity is related to GATA transcription factors (such as BZP2, GAT5, GAT6), which may regulate radiation response through nitrogen metabolism [46]. These examples show that plant transcription factors are often involved in the regulation of oxidative stress, DNA repair, and secondary metabolism. The AP2/ERF transcription factor family is widely involved in abiotic stress in plants [47], and has also been reported to be involved in oxidative defense and other processes [48], but whether or not it is related to radiation response is not explicitly mentioned. This study is the first to directly link the transcription factor OsERF110 of the AP2/ERF family with the radiation response of plants (especially germ cells), breaking the traditional understanding that the family is mainly involved in abiotic stresses such as drought and salt damage. This transcription factor participates in regulating the response mechanism of rice pollen to heavy ion irradiation by upregulating the expression of genes related to DNA damage repair, stimulus response, redox, and synthesis metabolism of various organic compounds, filling the gap in the research field regarding the correlation between AP2/ERF and radiation response.

## 5. Conclusions

In this study, the regression equation of dose survival rate of heavy ion radiation was constructed, and through the immunofluorescence experiment, it was found that DSBs induced by radiation could be repaired quickly, but the repair of complex damage required more time. RNA-seq analysis showed that there were significant differences in gene expression patterns at different time points. A total of 5,556 DEGs were screened out, and the number of DEGs decreased with time. At 0–1 h, DEGs were mainly involved in stress response (such as reactive oxygen species, temperature), protein folding, DNA repair, and other damage response processes; at 6 h, the cells turned to normal metabolism processes such as organic synthesis (e.g., lipid) and enzyme and protein activities. Combined with WGCNA and trend analysis, the key transcription factor OsERF110 was identified in response to heavy ion irradiation, which acts on the nucleus and membrane. A total of 45,680 OsERF110 binding peaks were identified by DAP-seq in the whole genome. Combined with RNA-seq, 62 OsERF110 target genes were further screened. These target genes were involved in DNA repair, stress response, redox, metabolic regulation, and other processes, forming the OsERF110 mediated radiation response regulatory network. In conclusion, this study systematically analyzed the dynamic process of DNA damage and repair in rice pollen irradiated by heavy ions and identified the core regulatory role of transcription factor OsERF110. The results of this study provide a new target for rice mutation breeding and lay a theoretical foundation for radiation biology research.

## Figures and Tables

**Figure 1 biology-14-01218-f001:**
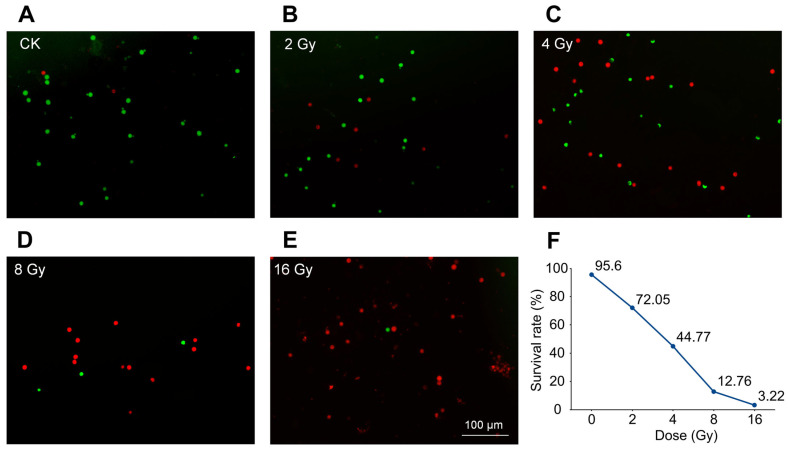
Pollen cell survival rate under different heavy ion irradiation doses. (**A**–**E**) The pollen survival rates at different doses, i.e., CK (0 Gy), 2 Gy, 4 Gy, 8 Gy, and 16 Gy, in which green indicates living cells, and red shows dead cells. (**F**) Dose survival regression curve.

**Figure 2 biology-14-01218-f002:**
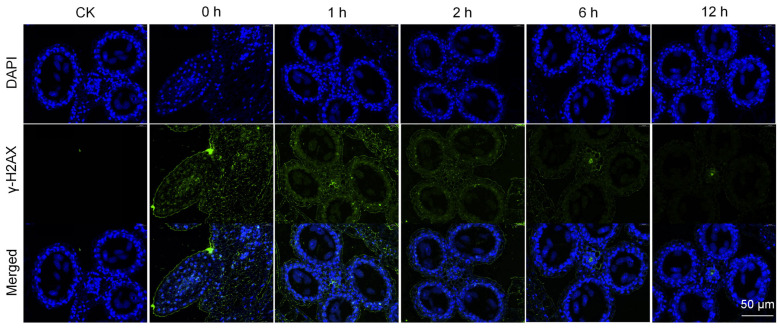
DSB repair at different time points after irradiation. The blue fluorescent dots represent the positions of the cell nucleus, while the green fluorescent dots represent DSBs.

**Figure 3 biology-14-01218-f003:**
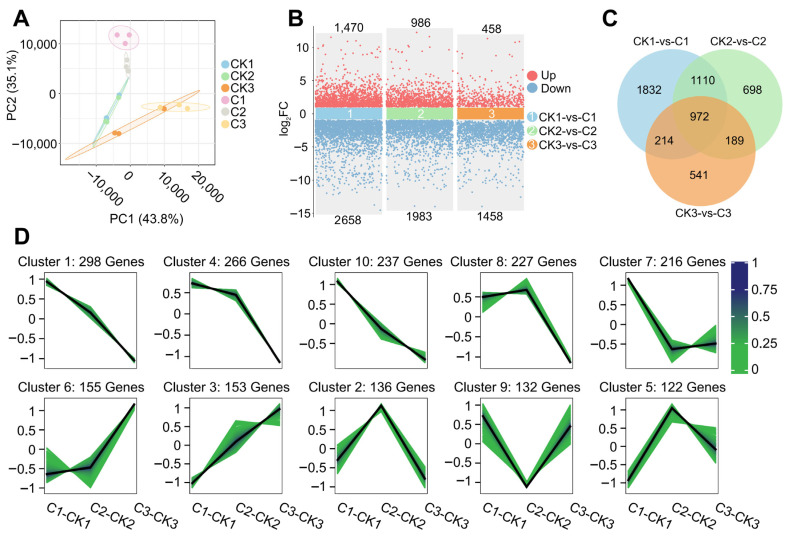
Gene expression differences at different time points after irradiation. (**A**) PCA of 18 transcriptome samples. The horizontal axis represents the first principal component; the percentage in parentheses represents the contribution of the first principal component to sample differences. The vertical axis represents the second principal component, and the percentage in parentheses represents the contribution of the second principal component to sample differences. (**B**) Number of DEGs at three sampling time points. Each scatter represents a gene, with those above indicating upregulated genes and those below indicating downregulated genes. The number represents the number of DEGs in each group. (**C**) The intersection of DEGs at three sampling time points. (**D**) Trend analysis of DEGs. The black line represents the target trend line, and each green line represents a gene; color represents the membership value of the genes.

**Figure 4 biology-14-01218-f004:**
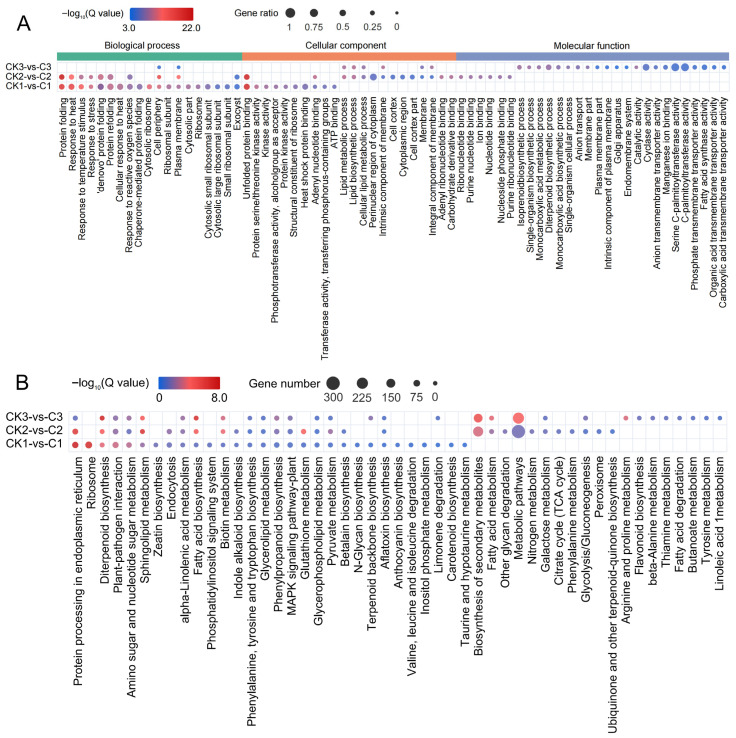
GO and KEGG enrichment of DEGs. (**A**) GO enrichment of DEGs: the top 10 GO terms of −log10 (Q value) of each sample are obtained; the size of the circle represents the number of genes, and the color represents the significance level. (**B**) KEGG enrichment of DEGs: the top 10 pathways of −log10 (Q value) of each sample are obtained. The size of the circle represents the number of genes, and the color represents the significance level.

**Figure 5 biology-14-01218-f005:**
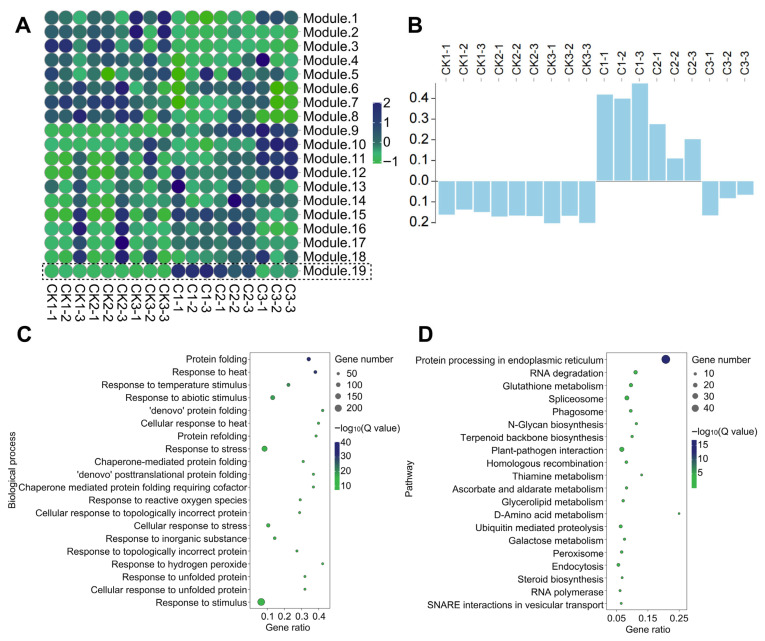
WGCNA of irradiation response gene. (**A**) WGCNA module. The color indicates the gene expression in the module. (**B**) The expression pattern of genes in module 19 among samples. (**C**) GO enrichment of module 19 genes (the top 20 biological processes of −log10 (Q value)); the circle size indicates the number of genes, and the color indicates the significance level. (**D**) KEGG enrichment of module 19 genes (the top 20 pathways of −log_10_ (Q value)); the size of the circle indicates the number of genes, and the color indicates the significance level.

**Figure 6 biology-14-01218-f006:**
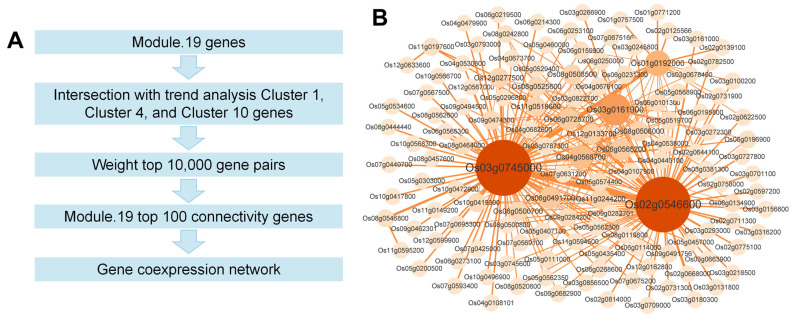
Identification of key genes in response to heavy ion irradiation. (**A**) Key gene identification process in response to heavy ion irradiation. (**B**) Gene coexpression network; the color and circle size represent the number of genes regulated, and the line thickness represents the weight value of gene pairs.

**Figure 7 biology-14-01218-f007:**
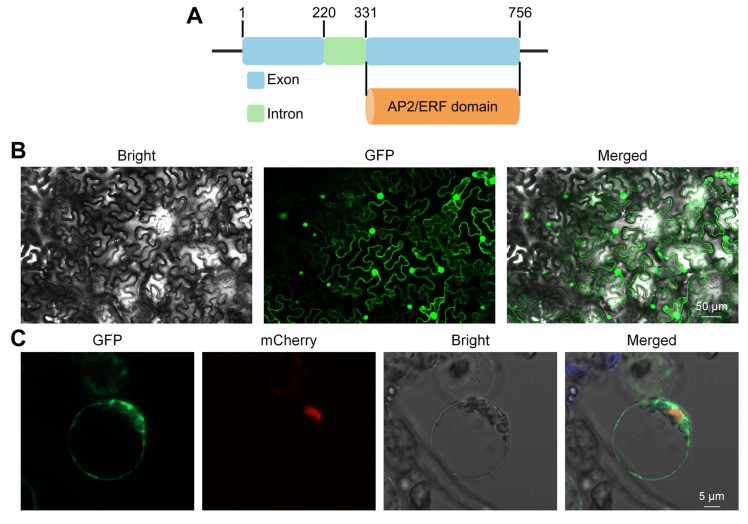
Structure and subcellular localization of *OsERF110*. (**A**) The structure of *OsERF110*. (**B**) Transient expression analysis of tobacco mesophyll cells. The green fluorescent position is the target transcription factor expression site. (**C**) Transient expression analysis of rice protoplasts. The green fluorescence indicates the expression location of the target transcription factor and GFP fusion protein, while the red fluorescence indicates the nuclear location.

**Figure 8 biology-14-01218-f008:**
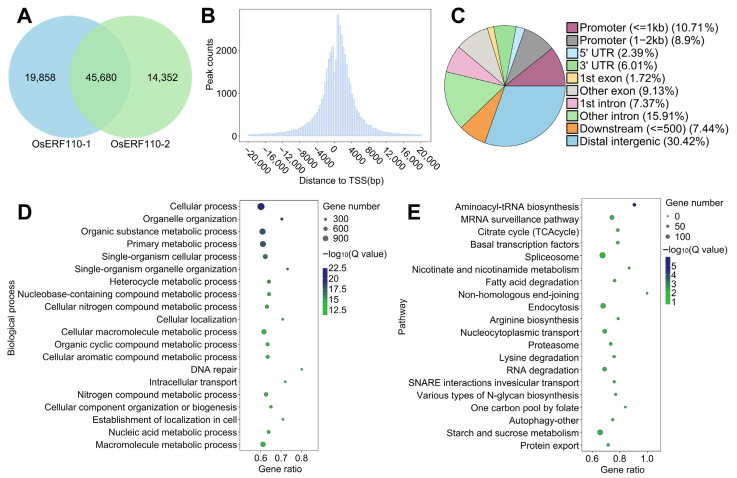
DAP-seq analysis of OsERF110. (**A**) The number of peaks in two biological replicates. (**B**) The distance distribution of peaks relative to TSS. (**C**) The distribution of peaks in the genome. (**D**) Peaks are located in the GO enrichment of target genes in the promoter region and the top 20 biological processes of −log10 (Q value). The size of the circle indicates the number of genes, and the color indicates the significance level. (**E**) Peaks are located in the KEGG enrichment of target genes in the promoter region, with the top 20 pathways of −log10 (Q value). The size of the circle indicates the number of genes, and the color indicates the significance level.

**Figure 9 biology-14-01218-f009:**
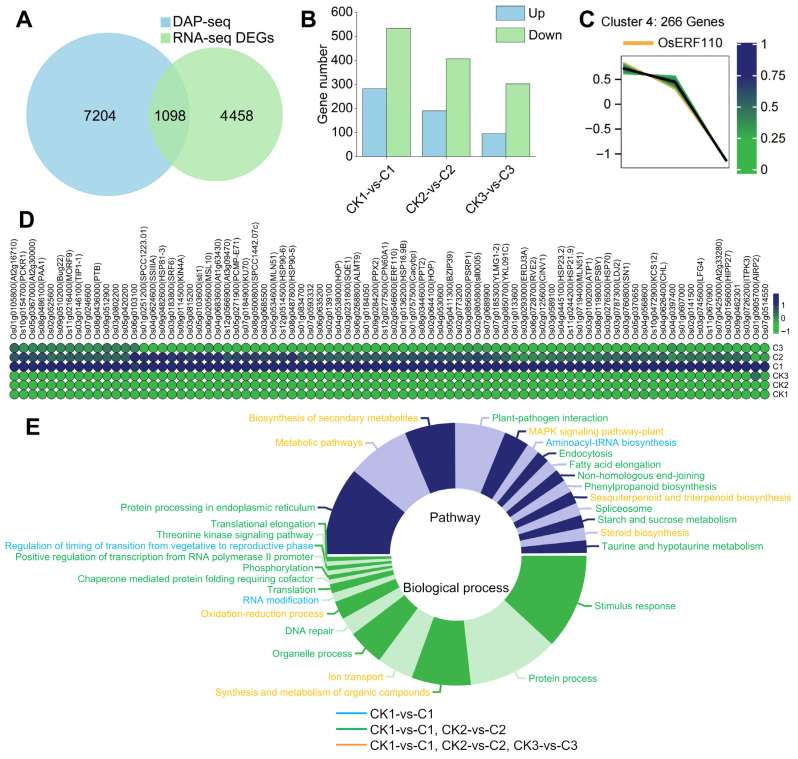
Analysis of OsERF110 regulatory network. (**A**) DAP-seq target genes intersected with RNA-seq DEGs. (**B**) The number and expression of intersection genes in different samples. (**C**) Trend analysis of *OsERF110*. The color represents the membership value. (**D**) The expression heat map of 62 target genes, and the color indicates the level of expression. (**E**) The functional analysis of 62 target genes; the upper half of the circle indicates the pathway, and the lower half of the circle indicates the biological process. The proportion of the circle graph indicates the number of genes in the pathway or biological process, and the color indicates the differential expression of genes in the sample.

## Data Availability

All data are available in the manuscript and Appendix A.

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
