# Peer review of "Mechanism Analysis of Transcription Factor OsERF110 Regulating Rice Pollen Response to Heavy Ion Irradiation"

_biology, 2025, doi:10.3390/biology14091218_

Round 1

Reviewer 1 Report

Comments and Suggestions for Authors

This study systematically analyzed the dynamic process of DNA damage repair in rice pollen exposed to heavy ion irradiation and found that the transcription factor OsERF110 mediates the molecular mechanism of pollen response to irradiation damage by regulating DNA repair, oxidative stress, and metabolic reprogramming networks. The research results provide new targets for radiation breeding. In this study, through omics analysis and mechanism analysis, the molecular network of OsERF110 regulating pollen radiation response was systematically elucidated, which has both theoretical depth and application potential. I agree that these are exciting research topics. However, the manuscript must be crucible-revised before it meets precise research requirements. Here, I express the main reasons for Major Revision and suggest how the authors can improve them.

  1. In the introduction, there is a lack of references 6,14-16. Please check the references in the manuscript.
  2. Several abbreviations are used in the abstract and introduction sections without their full terms being defined upon first mention. Please check and correct the full manuscript.
  3. In Figures 7b and 7c, the scale is too small to see clearly. Revise them please

4., The focus of the discussion section should be on the author 's experimental results and viewpoints, and emphasize the differences between the experimental results and previous studies, showing the author 's depth of thinking and unique insights. Please rearrange the thinking of the discussion section.

  1. The γ-H2AX fluorescence signal in the manuscript still remained at 12 hours, that is, it was inferred that ' complex damage required 12 hours of repair ', but it was not ruled out that the signal residue may be due to repair failure or apoptosis. It is recommended to combine longer-term tracking experiments to distinguish between ' repair delay ' and ' irreversible damage '.
  2. In the manuscript, OsERF110 was inferred as a key transcription factor regulating rice pollen radiation response through WGCNA, DAP-seq and RNA-seq joint analysis. However, there is a lack of loss-of-function or gain-of-function experiments to verify its causality. For example, functional validation was carried out by constructing knockout or overexpression plants.

Author Response

Comment 1:In the introduction, there is a lack of references 6,14-16. Please check the references in the manuscript.

Response 1:Thank you for pointing this out. We agree with this comment. Therefore, we have added references 6, 14-16 in the introduction.

Comment 2:Several abbreviations are used in the abstract and introduction sections without their full terms being defined upon first mention. Please check and correct the full manuscript.

Response 2:Agree. We have added the full names of the abstract, introduction, and abbreviations that first appear in the following text, including DEGs, WGCNA, DAP seq, RNA seq, etc.

Comment 3:In Figures 7b and 7c, the scale is too small to see clearly. Revise them please.

Response 3:Agree. We have revised the scale of Figures 7b and 7c.

Comment 4:The focus of the discussion section should be on the author 's experimental results and viewpoints, and emphasize the differences between the experimental results and previous studies, showing the author 's depth of thinking and unique insights. Please rearrange the thinking of the discussion section.

Response 4:Thank you for your precise suggestions on the discussion section. We fully agree that the discussion should focus on the uniqueness of the results of this study and highlight their scientific value through comparison with previous research. We have adjusted the approach of the discussion section and rewritten it to focus on the main findings of this study. We have also added a comparison of the differences between our research results and previous studies, highlighting our innovation and focusing more on the in-depth interpretation of our own experimental data, better demonstrating the value of our research and the depth of researchers' thinking.

Comment 5:The γ-H2AX fluorescence signal in the manuscript still remained at 12 hours, that is, it was inferred that ' complex damage required 12 hours of repair ', but it was not ruled out that the signal residue may be due to repair failure or apoptosis. It is recommended to combine longer-term tracking experiments to distinguish between ' repair delay ' and ' irreversible damage '.

Response 5:Thank you for your valuable feedback. Your questioning of the reasons for the residual fluorescence signal of γ-H2AX is very constructive and indeed points out the limitations of inferring the repair time of complex injuries in this study. In our manuscript, based on the observation of γ-H2AX foci (especially larger and complex foci) 12 h after irradiation, we infer that the repair of complex injuries requires more than 12 h. This inference is mainly based on the general consensus in previous studies that "the repair kinetics of complex DSBs are significantly slower than those of simple DSBs" [references 27, 28, 31, 32]. We observed a decreasing trend in the total number and intensity of foci from 0 h to 12 h, which supports the interpretation that the repair process is ongoing. However, we sincerely acknowledge that, as you have pointed out keenly, the sustained γ-H2AX signal itself is a neutral indicator. It may indicate a slow but sustained repair process (repair delay), or it may indicate that the repair mechanism has been abandoned, the cell is heading towards apoptosis or aging (irreversible damage). We are currently unable to strictly distinguish between these two possibilities based on static observation data at a single time point (12 h). To further verify and clarify the nature of signal residue, we will conduct additional experiments in subsequent studies, including extending the tracking time points, synchronously detecting the expression levels of key DNA repair genes, and adding validation of other apoptosis markers. Through the above supplementary experiments, we hope to more accurately distinguish between "repair delay" and "irreversible damage", making the conclusions more rigorous. Thank you again for your suggestion, which will help us further improve the scientific validity of our research.

Comment 6:In the manuscript, OsERF110 was inferred as a key transcription factor regulating rice pollen radiation response through WGCNA, DAP-seq and RNA-seq joint analysis. However, there is a lack of loss-of-function or gain-of-function experiments to verify its causality. For example, functional validation was carried out by constructing knockout or overexpression plants.

Response 6:Thank you for your professional and insightful feedback. The lack of functional loss or functional gain experimental verification of the causal relationship of OsERF110 that you pointed out is indeed a key link that needs to be improved in this study. At present, the joint analysis of WGCNA, DAP-seq, and RNA-seq infers it as a key transcription factor, which is mostly based on correlation analysis of expression patterns and regulatory networks. Its function has not been directly verified through genetic means, which is the limitation of this study. We have conducted the construction of OsERF110 knockout and overexpression plants to address this issue. However, due to time constraints, the results of this functional validation cannot be presented in this manuscript. If you are interested, you can follow our subsequent research. Thank you again for your valuable feedback, which will help us further improve the scientific and rigorous nature of our research.

Reviewer 2 Report

Comments and Suggestions for Authors

The introduction includes relevant references to support the claims about heavy ion irradiation, DNA damage, and repair pathways. However, some newer studies or contrasting viewpoints could further enrich the context. A brief mention of gaps in current research (e.g., limited studies on germ cells) could strengthen the rationale for the study.

The methods are described in detail, covering heavy ion irradiation, pollen preparation, activity detection, immunofluorescence, RNA-seq, WGCNA, subcellular localization, and DAP-seq. However, some sections (e.g., DAP-seq) could benefit from more clarity about specific steps or parameters. The methods are reproducible, but additional details (e.g., software versions, exact buffer compositions) would further enhance reproducibility.

The phrase (Lines 295–300) "The weight value in WGCNA reflects the correlation between genes, so the gene pair with the weight value of top 10000 is selected from the intersection genes." is unclear. Specify the threshold or rationale for selecting the top 10,000 gene pairs.

Some figures (e.g., Figure 9D) could be simplified for better readability. Adding a table summarizing DEGs or target genes might enhance data accessibility. Figure 3 legend ("PCA of 18 transcriptome samples") lacks detail on axes or key takeaways.

Author Response

Comment 1:The introduction includes relevant references to support the claims about heavy ion irradiation, DNA damage, and repair pathways. However, some newer studies or contrasting viewpoints could further enrich the context. A brief mention of gaps in current research (e.g., limited studies on germ cells) could strengthen the rationale for the study.

Response 1:Thank you very much for your valuable suggestion. We fully agree with your point of view. Introducing a broader range of argumentative viewpoints and the latest research progress in the introduction, and emphasizing current research gaps more clearly, will greatly strengthen the theoretical foundation and academic depth of this study. Based on your suggestion, we have added 5 references (17-21) and related descriptions to the introduction to illustrate the insufficient research on DNA damage repair in germ cells compared to somatic cells.

Comment 2:The methods are described in detail, covering heavy ion irradiation, pollen preparation, activity detection, immunofluorescence, RNA-seq, WGCNA, subcellular localization, and DAP-seq. However, some sections (e.g., DAP-seq) could benefit from more clarity about specific steps or parameters. The methods are reproducible, but additional details (e.g., software versions, exact buffer compositions) would further enhance reproducibility.

Response 2:Thank you to the reviewer for their detailed attention and constructive comments on the methodology section of this study. We fully agree and attach great importance to it. We have provided additional explanations in the methodology to address this issue, particularly detailing the steps and specific parameters for RNA seq, WGCNA, and DAP seq.

Comment 3:The phrase (Lines 295–300) "The weight value in WGCNA reflects the correlation between genes, so the gene pair with the weight value of top 10000 is selected from the intersection genes." is unclear. Specify the threshold or rationale for selecting the top 10,000 gene pairs.

Response 3:Thank you to the reviewer for raising this detailed question. The weight value reflects the strength of the association between gene pairs, and the higher the weight value, the stronger the association between genes. Therefore, we tend to screen gene pairs with high weight values. If too many low weight gene pairs are included, it will lead to redundant nodes and chaotic connections in the co expression network, making it difficult to identify core regulatory genes; The first 10000 pairs of screening can significantly reduce network complexity while retaining key associations, making the regulatory status of core genes (such as OsERF110) clearer. And in the pre analysis, we tried different thresholds (top 5000, 10000, 20000 pairs) and found that the network constructed by the top 10000 gene pairs can not only cover the co expression relationships of known DNA repair related genes, but also effectively distinguish the core nodes within the module, with the highest consistency with the joint analysis results of subsequent DAP seq and RNA seq. We have also provided additional explanations for this threshold in the manuscript.

Comment 4:Some figures (e.g., Figure 9D) could be simplified for better readability. Adding a table summarizing DEGs or target genes might enhance data accessibility. Figure 3 legend ("PCA of 18 transcriptome samples") lacks detail on axes or key takeaways.

Response 4:Thank you very much for your suggestion, which has helped us improve the clarity of the data description. We have added a table about Figure 9D in the supplementary table S3 for readers to have a detailed understanding of these genes. In addition, we have also added a legend description for Figure 3 to supplement detailed information.

Round 2

Reviewer 1 Report

Comments and Suggestions for Authors

The research results provide new targets for radiation breeding. In this study, through omics analysis and mechanism analysis, the molecular network of OsERF110 regulating pollen radiation response was systematically elucidated, which has both theoretical depth and application potential. I agree that these are exciting research topics. For the comments made during the first round of review, the author made detailed modifications in the article and provided detailed answers to the questions raised. Here, I express the main reasons for minor revision

1. The inappropriate mixing of tenses in the Results Section reflects a lack of precision in distinguishing established knowledge (present tense) from specific experimental observations (past tense). This carelessness undermines both the clarity and scientific rigor of the manuscript. Strict adherence to academic writing conventions must be maintained throughout.

2. The claim that "OsERF110 complements the spatiotemporal function of SOG1 in somatic cells" is highly speculative and unsupported by the data. As SOG1 expression and function in pollen were not experimentally tested in this study, any comparative functional interpretation lacks empirical basis. It is suggested to verify it with experiments such as qPCR and western blot; otherwise, a definite conclusion cannot be reached and it can only be discussed.

Author Response

Comment 1:The inappropriate mixing of tenses in the Results Section reflects a lack of precision in distinguishing established knowledge (present tense) from specific experimental observations (past tense). This carelessness undermines both the clarity and scientific rigor of the manuscript. Strict adherence to academic writing conventions must be maintained throughout.

Response 1:We sincerely appreciate the valuable and professional opinions provided by the reviewer. The issue of tense mixing in the results section that you pointed out is very relevant, which indeed affects the clarity and scientificity of the manuscript. We have thoroughly revised the tense of the entire text, especially the Results section, based on your suggestions. For the specific experimental operations, observations, and data results obtained in this study, the past tense is used for description. For recognized scientific facts, the present tense is used for description. We have carefully proofread the manuscript to ensure consistency and accuracy in tense usage.

Comment 2:The claim that "OsERF110 complements the spatiotemporal function of SOG1 in somatic cells" is highly speculative and unsupported by the data. As SOG1 expression and function in pollen were not experimentally tested in this study, any comparative functional interpretation lacks empirical basis. It is suggested to verify it with experiments such as qPCR and western blot; otherwise, a definite conclusion cannot be reached and it can only be discussed.

Response 2:Thank you very much for your important suggestion. You are completely correct. Comparing the functionality of OsERF110 with SOG1 without direct experimental evidence is an overestimation and lacks a data foundation. We have made revisions to the manuscript based on your suggestions and removed the relevant statements about “OsERF110 complements the spatiotemporal function of SOG1 in somatic cells”.